# OpenReview forum: "Certified Neural PDE Solvers: Constructive Verification of Physics-Informed Networks via Dependent Type Theory"
_mathai.club/MathAI/2026/Conference — MathAI 2026 Conference Submission_

### Official Review · Reviewer_B7aQ · 2026-03-11
**Promising idea but a bit early-stage realization**

**Rating:** 5
**Confidence:** 5

**Review:**

### Summary

This paper proposes CertPINN, a framework that combines physics-informed neural networks with dependent type theory (Lean 4) to produce formally verified PDE solutions. The core idea—encoding both the solution and a proof of its error bound as a dependent type—is novel and addresses a genuine limitation of standard PINNs: the lack of guarantees beyond point-wise loss minimization. The authors present a training algorithm, convergence analysis, a posteriori error bounds, and experimental results on five benchmark PDEs.

### Major Concerns

1. Computational cost claims are implausible (Section 6.2)

Table 2 reports interval arithmetic taking 2–8 **milliseconds** per epoch and Lean verification completing in 0.4–1.2 **seconds** for 2D+time PDEs with neural networks of moderate size (e.g., 4×64 for Burgers). These numbers are not credible:

- Interval propagation through a neural network for a single box requires significantly more computation than a forward pass. For a 100×100 grid (10,000 boxes), 2 ms total would mean **0.2 microseconds per box**—impossible even for trivial networks.
- Lean 4 proof checking for a certification involving thousands of boxes and a network with thousands of parameters would take orders of magnitude longer than 0.4 seconds, especially given the complexity of formalizing real analysis and PDE theory.
- Without details on the number of boxes, network size, or what "per epoch" actually measures, these numbers cannot be evaluated. They appear optimized to make the overhead seem negligible (3–7%) while hiding true computational cost.

2. The Lean 4 verification pipeline is not provided (missing appendix)

The algorithm calls `VerifyInLean` (line 303) and references an appendix, but no appendix is included. This is a critical omission:

- The core contribution - machine-checkable certificates - cannot be assessed or reproduced.
- It is unclear whether the Lean formalization actually exists, whether it covers the necessary real analysis (Sobolev spaces, PDE operators, interval arithmetic), or whether the proof terms are constructed automatically or manually.
- The paper's claim of "growing libraries for analysis and PDEs" in Lean 4 is optimistic; formalizing the required mathematics is a massive undertaking that is not demonstrated.

3. No guidance for uncertified solutions

The algorithm simply continues training if certification fails (lines 326–328). There is no discussion of:
- How to interpret per‑box residual bounds to guide adaptive sampling or architecture changes.
- What to do if the network *never* certifies (e.g., due to insufficient capacity or poor conditioning).
- A stopping criterion beyond "keep training forever."

This reduces certification to a binary pass/fail with no actionable feedback—a missed opportunity given that interval arithmetic provides rich diagnostic information.

4. Convergence proof relies on strong assumptions

Theorem 8 claims convergence in $O(\log(1/\epsilon))$ iterations, which is unusually fast. The proof sketch cites "online convex optimization analysis" but:
- Neural network loss landscapes are non-convex, so convex analysis does not apply.
- Assumption (A2) requires the differential operator $\mathcal{F}$ to be Lipschitz it is too strong for nonlinear PDEs like Navier-Stokes.
- The approximation error $\delta_{\mathrm{approx}}$ is assumed without discussion of how it is estimated or bounded in practice.

The theorem gives an appearance of rigor but rests on assumptions that are not satisfied in realistic settings.

### Minor Issues

- Equation (3) contains basic type errors (unbound variable `ε`, mis‑typed `u`, unused parameter `f`). While later definitions are better, this sloppiness undermines confidence in the authors' grasp of dependent type theory.
- The interval arithmetic description (Section 5.2) is too brief: how are boxes chosen? How is the dependency problem handled? How are activation function ranges computed rigorously?
- The Burgers "silent failure" example is compelling, but it's unclear whether CertPINN actually fixed the problem or just detected it.

### Potential

Despite these issues, the **core idea** is valuable. Combining formal verification with PINNs could indeed lead to trustworthy scientific ML. The paper identifies a real problem and proposes a principled solution. If the authors provided:
- A working Lean implementation (with code),
- Realistic timing measurements on standard hardware,
- A clear explanation of how certification failures guide training,
- A more modest convergence claim,

this could be a strong contribution. In its current form, however, it reads as an ambitious proposal with insufficient evidence.

---

### Official Review · Reviewer_BaPM · 2026-03-12
**A clear neural PDE verification method with theory, proof sketches, and experiments**

**Rating:** 6
**Confidence:** 4

**Review:**

The paper addresses the problem of formally verifying neural-network-based solutions of partial differential equations. The authors propose CertPINN, a framework that combines physics-informed neural networks with dependent type theory, interval arithmetic, and Lean 4 verification in order to produce machine-checkable certificates that the learned solution satisfies the governing PDE and boundary conditions within a prescribed tolerance. The paper includes a type-theoretic formalization of weak PDE solutions, a certified training objective, a convergence statement under explicit assumptions, an a posteriori error estimator, and experiments on five benchmark PDEs
Among the strengths of the paper is the fact that it is not limited to a high-level conceptual proposal, but attempts to connect method construction, theoretical analysis, and practical validation in a single framework. The paper gives a clear description of the proposed method, including the certified solution type, the certified loss, the interval-arithmetic-based verification component, and the Lean 4 verification pipeline. A further strong point is the presence of explicit assumptions under which the main convergence claim is stated, including well-posedness of the PDE, Lipschitz continuity of the operator, approximation capacity of the neural network class, and optimization conditions. This makes the theoretical part more structured and easier to interpret than in many purely empirical PINN papers. The paper is also strengthened by practical examples across several classes of PDEs, and by experiments indicating that the additional computational cost of certification remains moderate while the resulting solution is substantially more valuable than a standard PINN solution because it comes with a formal correctness guarantee rather than only empirical evidence. The Burgers case study is particularly useful, since it illustrates that the proposed approach can detect failure modes that a standard PINN may miss despite a small residual loss.
At the same time, the paper has some weaknesses. Part of the theoretical justification is presented only in the form of proof sketches rather than full proofs, which somewhat reduces the level of rigor of the presentation. In addition, although the experiments are useful and illustrative, the empirical section is relatively limited.
As recommendations for improvement, the authors could expand the proof sketches into more complete arguments in order to strengthen the theoretical rigor of the paper. In addition, the empirical section could be extended with a larger number of experiments and a more detailed evaluation. Overall, the paper presents an interesting and potentially important direction at the intersection of scientific machine learning and formal verification, and the proposed idea appears both original and practically meaningful.

---

### Decision · Program_Chairs · 2026-03-20

**Decision:**

Reject

**Comment:**

After careful evaluation by the Program Committee, we regret to inform you that your submission has not been accepted for presentation at MathAI 2026.

All submissions underwent a rigorous two-stage review process. Unfortunately, the reviewers identified one or more of the following concerns with your paper:

- Insufficient mathematical rigor or novelty relative to the existing body of work in the field;
- Presentation of results that substantially overlap with or rephrase previously published findings without clear original contribution;
- Significant issues with technical quality, including but not limited to broken or non-existent references, unsupported claims, or methodological gaps;
- Indications that the manuscript may have been generated with the assistance of large language models without substantial original intellectual contribution by the authors.

We received a large number of submissions this year, and the selection process was highly competitive. We encourage you to carefully consider the reviewers’ feedback (available through OpenReview), revise your work accordingly, and consider submitting an improved version to a future edition of MathAI or to another appropriate venue.

We appreciate your interest in MathAI and hope you will continue to engage with the conference community.

With kind regards,

MathAI 2026 Program Committee
International Conference on Mathematics of Artificial Intelligence
https://mathai.club
OpenReview: https://openreview.net/group?id=mathai.club/MathAI/2026/Conference
MathAI Telegram: https://t.me/MathAI_club
IAIC International AI Committee: https://t.me/iaic_world
Email: mathai.club@yandex.ru